# Coenzyme Q10 encapsulated in micelles ameliorates osteoarthritis by inhibiting inflammatory cell death

Hyun Sik Na[1,2]☯, Jin Seok Woo[1]☯, Ju Hwan Kim[3]☯, Jeong Su Lee[1,2], In Gyu Um[1,2], Keun-Hyung Cho[1,2], Ga Hyeon Kim[3], Mi-La Cho[1,2,4]*, Sang J. Chung[5]*, Sung-Hwan Park[1,6]*

1 The Rheumatism Research Center, Catholic Research Institute of Medical Science, College of Medicine, The Catholic University of Korea, Seoul, Korea, 2 Department of Biomedicine & Health Sciences, College of Medicine, The Catholic University of Korea, Seoul, Korea, 3 AbTis Co. Ltd., Suwon, Korea, 4 Department of Medical Lifesciences, College of Medicine, The Catholic University of Korea, Seoul, Korea, 5 Department of Biopharmaceutical Convergence, School of Pharmacy, Sungkyunkwankwan University, Suwon, Korea, 6 Division of Rheumatology, Department of Internal Medicine, Seoul St. Mary's Hospital, College of Medicine, The Catholic University of Korea, Seoul, Korea

☯ These authors contributed equally to this work.
* iammila@catholic.ac.kr (MLC); sjchung@skku.edu (SJC); rapark@catholic.ac.kr (SHP)

**Data Availability Statement:** All relevant data are within the paper and its Supporting information files.

## Abstract

### Background

Osteoarthritis (OA) is the most common degenerative joint disease and is characterized by breakdown of joint cartilage. Coenzyme Q10 (CoQ10) exerts diverse biological effects on bone and cartilage; observational studies have suggested that CoQ10 may slow OA progression and inflammation. However, any effect of CoQ10 on OA remains unclear. Here, we investigated the therapeutic utility of CoQ10-micelles.

### Methods

Seven-week-old male Wistar rats were injected with monosodium iodoacetate (MIA) to induce OA. CoQ10-micelles were administered orally to MIA-induced OA rats; celecoxib served as the positive control. Pain, tissue destruction, and inflammation were measured. The expression levels of catabolic and inflammatory cell death markers were assayed in CoQ10-micelle-treated chondrocytes.

### Results

Oral supplementation with CoQ10-micelles attenuated OA symptoms remarkably, including pain, tissue destruction, and inflammation. The expression levels of the inflammatory cytokines IL-1β, IL-6, and MMP-13, and of the inflammatory cell death markers RIP1, RIP3, and pMLKL in synovial tissues were significantly reduced by CoQ10-micelle supplementation, suggesting that CoQ10-micelles might attenuate the synovitis of OA. CoQ10-micelle addition to cultured OA chondrocytes reduced the expression levels of catabolic and inflammatory cell death markers.

**Funding:** Initials of the authors who received each award: S.H.P. Grant numbers awarded to each author:HI20C1496 The full name of each funder: the Ministry of Health & Welfare, Republic of Korea URL of each funder website: https://www.mohw.go.kr/eng/ The funders had no role in study design, data collection and analysis, decision to publish, or preparation of the manuscript.

**Competing interests:** The authors have declared that no competing interests exist.

## Conclusions

CoQ10-micelles might usefully treat OA.

## Introduction

Osteoarthritis (OA) is a common degenerative joint disease associated with aging. OA compromises the quality-of-life and causes disabilities [1, 2]. OA is characterized by infiltration of immune cells into cartilage, progressive cartilage destruction, and chronic pain [3–5]. The risk factors for OA progression and development include joint injury, age, obesity, and gender [6–8]. Cytokines and chemokines (inflammatory mediators) are produced in OA joint tissue, including the synovium and cartilage, triggering cartilage destruction [9]. Although OA does not affect life expectancy, it significantly reduces the quality-of-life. Emerging drug treatments focus on pain relief, not on slowing of OA progression.

Antioxidants play crucial roles as anti-inflammatories. Oxidative stress induces an inflammatory response that plays a major role in the pathogenesis of many diseases [10–12]. Oxidative stress reduction is important in terms of disease management. Coenzyme Q10 (CoQ10), also known as ubiquinone-10, is an oil-soluble substance that scavenges free radicals; it is thus an antioxidant [13]. Also, CoQ10 participates in energy production, aiding adenosine triphosphate synthesis in mitochondria [14–16]. One clinical study found that OA patients exhibited low levels of CoQ10 and that the extent of oxidative stress was negatively correlated with the CoQ10 level [17]. Supplementation with a combination of vitamin-$K_2$ and CoQ10 improved pain, stiffness, and the daily performance of OA patients [18]. Several studies have reported therapeutic effects of CoQ10 in patients with inflammatory disorders [19–21]. Li et al. showed that CoQ10 ameliorated IL-1β-induced inflammation by inhibiting the MAPK signaling pathway in a rat OA model [22]. We previously reported that CoQ10 ameliorated OA development and progression by regulating the levels of nitric oxide and inflammatory cytokines [21, 23].

Drugs encapsulated in micelles exhibit several advantages compared to non-encapsulated drugs, including 1) increased water solubility; 2) less toxicity; 3) enhanced membrane permeability; and, 4) drug protection from the external environment [24–26]. The therapeutic effects of CoQ10-micelles have been evaluated in models of various diseases, but not OA [27–29]. Here, we explored the effects of CoQ10-micelles on OA development and progression.

## Materials and methods

### Animals

Seven-week-old male Wistar rats were purchased from Central Laboratory Animal Inc. (Korea). All animal research procedures were conducted in accordance with the Laboratory Animals Welfare Act, the Guide for the Care and Use of Laboratory Animals, and the Guidelines and Policies for Rodent Experiments of the Institutional Animal Care and Use Committee (IACUC) of the School of Medicine, the Catholic University of Korea.

### Preparation of coenzyme Q10 (CoQ10)-micelles

Coenzyme Q10 (CoQ10), eicosapentaenoic acid (EPA), and dipotassium glycyrrhizinate were purchased from Kaneka Nutrients (USA), Phycoil Biotech Korea Inc. (Korea), and MAFCO Worldwide LLC (US) respectively. CoQ10, EPA, and dipotassium glycyrrhizinate were dispersed in hot water at 1 mg/mL, mixed, and homogenized (Multi-Purpose Homogenizer,

YSTRAL, Germany) at 10,000 rpm until the consistency was uniform. An M-110P microfluidizer processor (Microfluidics, US) was used to create nanoemulsions. The reservoir capacity is 1,500 mL and the maximum operating pressure 30,000 psi. The mixture (at 60°C) was poured into the fluidizer. Nanoemulsions were obtained after five cycles of exposure to 30,000 psi. Nanoemulsion sizes were evaluated in triplicate via dynamic light scattering (DLS) using a Zetasizer Nano ZS90 (Malvern Instruments Ltd., UK). Before measurement, each nanoemulsion was diluted in distilled water at 25°C. The DLS results were processed by the Zetasizer software. Nanoemulsion components were quantified via reverse-phase high-pressure liquid chromatography (Waters 1525, USA) (C18 column of pore size 5 μm, 4.6 x 250 mm in dimensions; the Waters Xbridge). A gradient system was used to elute dipotassium glycyrrhizinate and an isocratic system to elute CoQ10 and EPA. The injected volume was 10 μL at a flow rate of 1 mL/min; the UV detector operated at 220, 254, and 280 nm. For CoQ10 analysis, the isocratic elution fluid was 40% methanol and 60% isopropyl alcohol (both v/v). For dipotassium glycyrrhizinate and EPA, gradient elution employed deionized water with 0.1% (v/v) trifluoroacetic acid (TFA) and acetonitrile with 0.075% (v/v) TFA (0–90% acetonitrile over 30 min).

## Induction of osteoarthritis and treatment with CoQ10-micelles

Animals were randomly assigned to the treatment or control groups before the study commenced. After anesthetization with isoflurane, rats (n = 3 per group) were injected with 3 mg of monosodium iodoacetate (MIA; Sigma, USA), dissolved in 50 μL of saline via a 26.5-G needle inserted through the patellar ligament into the intra-articular space of the right knee. 3 days after induction, CoQ10-micelles (5 mg/kg), CoQ10 (5 mg/kg), micelle (5 mg/kg), or celecoxib (80 mg/kg) in saline or saline alone for control were administered orally once a day. Animals were monitored for 21 days.

## Assessment of pain and weight-bearing

Nociception in MIA-treated rats was tested using a dynamic plantar aesthesiometer (Ugo Basile, Italy). This is an automated version of the von Frey hair procedure that measures mechanical sensitivity. Each rat was placed on a metal mesh in an acrylic chamber of a temperature-controlled room (20–26°C) and rested for 10 min before the touch stimulator was positioned underneath the animal. An adjustable angled mirror was used to place the stimulating microfilament (0.5 mm in diameter) below the plantar surface of the hind paw. When the instrument was activated, the fine plastic monofilament advanced at a constant speed and touched the paw in the proximal metatarsal region. The filament exerted a gradually increasing force on the plantar surface, starting below the threshold of detection and increasing until the stimulus became painful, as indicated by the rat's withdrawal of its paw. The force required to elicit a paw-withdrawal reflex was recorded automatically and measured in g. A maximum force of 50 g and a ramp speed of 25 s were used for all aesthesiometer tests. Weight balance in MIA-treated rats was analyzed using an incapacitance meter (IITC Life Science, USA). The rats were allowed to acclimate for 5 min in an acrylic holder. After 5 min, two feet were fixed to the pad and the weight balance measured over 5 s. Three replicate measurements were made. The weights borne by the unguided and guided legs were determined and used to calculate weight-bearing percentages; we compared legs with and without OA.

## *In vivo* microcomputed tomography (micro-CT) imaging and analysis

Micro-CT imaging and analysis were performed using a benchtop, cone-beam animal scanner (mCT 35; SCANCO Medical, Switzerland). The parameters were 70 kVp and 141 μA; we used a 0.5-mm-thick aluminum filter. The pixel size was 8.0 μm and the rotation step 0.4°C. Cross-

sectional images were reconstructed using a filtered back-projection algorithm (NRecon software, Bruker Micro CT, Belgium). For each scan, a stack of 286 cross-sections was reconstructed at $2,000 \times 1,335$ pixels. Bone volume and surface were analyzed at the femur.

## Histopathological analysis

Knee joints and dorsal root ganglions were collected from each group 3 weeks post-MIA induction. The tissues were fixed in 10% (v/v) formalin, decalcified using Decalcifying Solution-Lite (Sigma, USA), and embedded in paraffin. Sections of thickness 4- to 5-μm were cut, dewaxed in xylene, dehydrated through an alcohol gradient, stained with safranin-O, and scored using the Osteoarthritis Research Society International (OARSI) and Mankin systems.

## Immunohistochemistry

Paraffin-embedded sections were incubated at 4˚C with the following primary monoclonal antibodies: Anti-IL-1β (1:400 dilution, nb600-633, Novus [USA]), anti-MMP-13 (1:300 dilution, ab39012, Abcam [USA]), anti-IL-6 (1:200 dilution, nb600-1131, Novus), anti-RIP1 (1:400 dilution, PA5-20811, Invitrogen [USA]), anti-RIP3 (1:200 dilution, Invitrogen), and anti-phospho-MLKL (1:200 dilution, Abcam). The samples were then incubated with appropriate secondary biotinylated antibodies, followed by 30-min incubation with a streptavidin-peroxidase complex. The reaction product was developed using the 3,3-diaminobenzidine chromogen (Dako, USA).

## Human articular chondrocyte differentiation

Human articular cartilage was acquired from patients undergoing replacement arthroplasty or joint replacement surgery and digested with 0.5 mg/mL hyaluronidase, 5 mg/mL protease type XIV, and 2 mg/mL collagenase type V. The resulting chondrocytes were incubated in Dulbecco's Modified Eagle's Medium (DMEM) with 10% (v/v) fetal bovine serum.

## RNA isolation, cDNA synthesis, and real-time quantitative PCR

RNA was extracted using the TRIzol reagent (Molecular Research Center Inc., USA). cDNA was synthesized using the Superscript Reverse Transcription System (TaKaRa, Japan), and quantitative real-time quantitative PCR was performed using a LightCycler FastStart DNA Master SYBR Green I kit (TaKaRa) according to the manufacturer's instructions. Expression values were normalized to that of β-actin. Primer sequences are listed in S1 Table in S1 File.

## Ethics approval and consent of participate

All experimental procedures were reviewed and approved by the Animal Research Ethics Committee of the Catholic University of Korea (approval no. 2020-0020-01). All experimental procedures were reviewed and approved by the Institutional Review Board of Seoul St. Mary's Hospital (approval no. UC14CNSI0150).

## Statistical analyses

Data are presented as means ± standard errors of the means (S.E.Ms.). All statistical analyses were performed using GraphPad Prism ver. 5 software for Windows (GraphPad Software, USA). Normally distributed continuous data were analyzed using the Bonferroni test. For all analyses, $p < 0.05$ was taken to indicate statistical significance.

## Results

### Preparation of coenzyme Q10-encapsulated micelles (CoQ10-micelles)

To generate CoQ10-micelles, we combined three substances (CoQ10, EPA, and dipotassium glycyrrhizinate) (Fig 1A, see the Methods). EPA, coenzyme Q10, and dipotassium glycyrrhizinate hydrate exhibited single peaks at retention times of 6.2, 6.9, and 15.9 min, respectively (Fig 1B). The concentrations were 0.24, 1.0, and 0.6 mg/mL respectively (S2 Table in S1 File). DLS revealed that the nanoemulsions were homogeneous with narrow size distributions around 100 nm (Fig 1C).

### Attenuation of OA progression by CoQ10-micelles

We administered CoQ10-micelles to MIA-induced OA rats; CoQ10 [21] and celecoxib [30] served as the positive control and micelle served as negative control. CoQ10-micelle exhibited improvements in paw withdrawal latency (PWL), the paw withdrawal threshold (PWT) (Fig 2A), and weight-bearing (Fig 2B).

### Protective effects of CoQ10-micelles on knee joints of MIA-induced OA rats

Micro-CT showed that the femora of OA rats were less damaged in the CoQ10-micelle than the vehicle group (Fig 3A and 3B). Safranin-O staining showed that cartilage destruction was

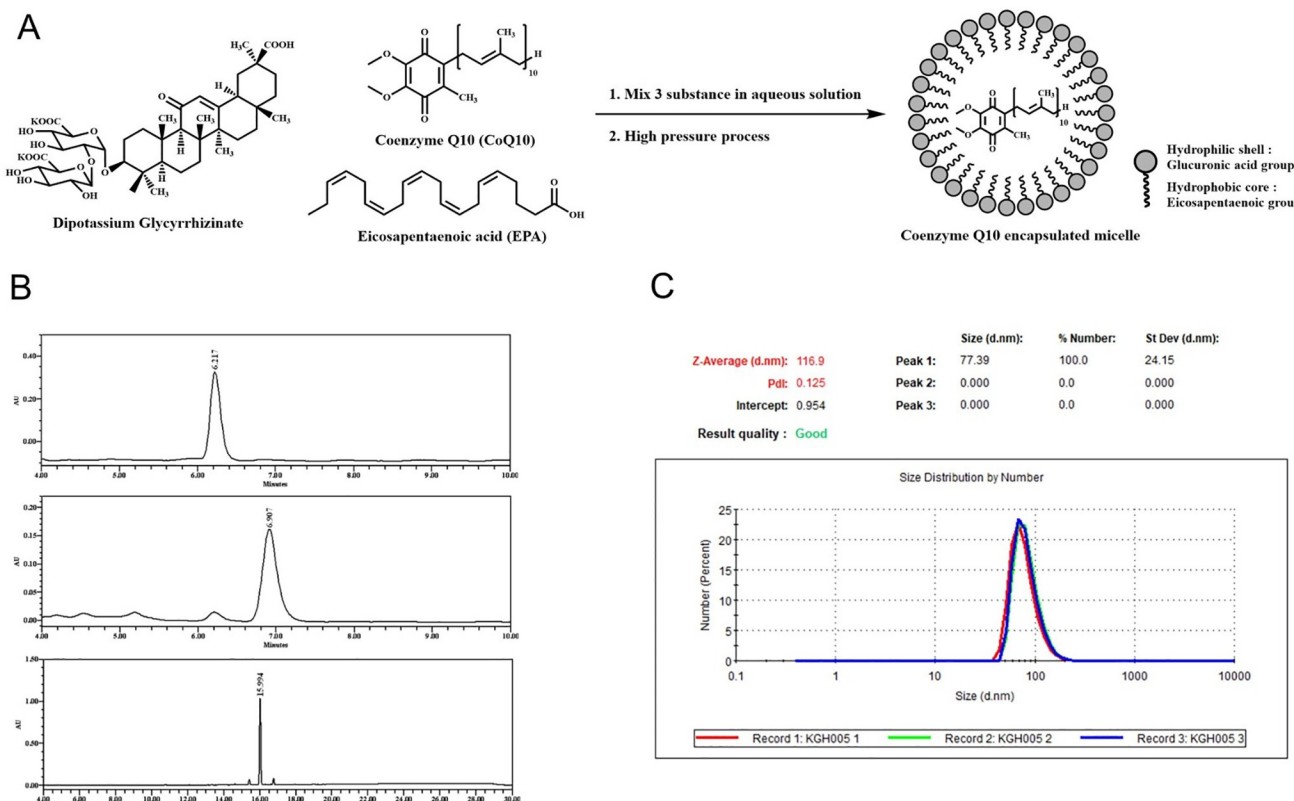

**Fig 1. Preparation of encapsulated CoQ10. (A)** Schematic showing preparation of CoQ10-micelles. **(B)** The HPLC profile of the nanoemulsion: EPA (top), CoQ10 (middle), and dipotassium glycyrrhizinate (bottom). **(C)** Nanoemulsion particle size distribution as revealed by dynamic light scattering (DLS).

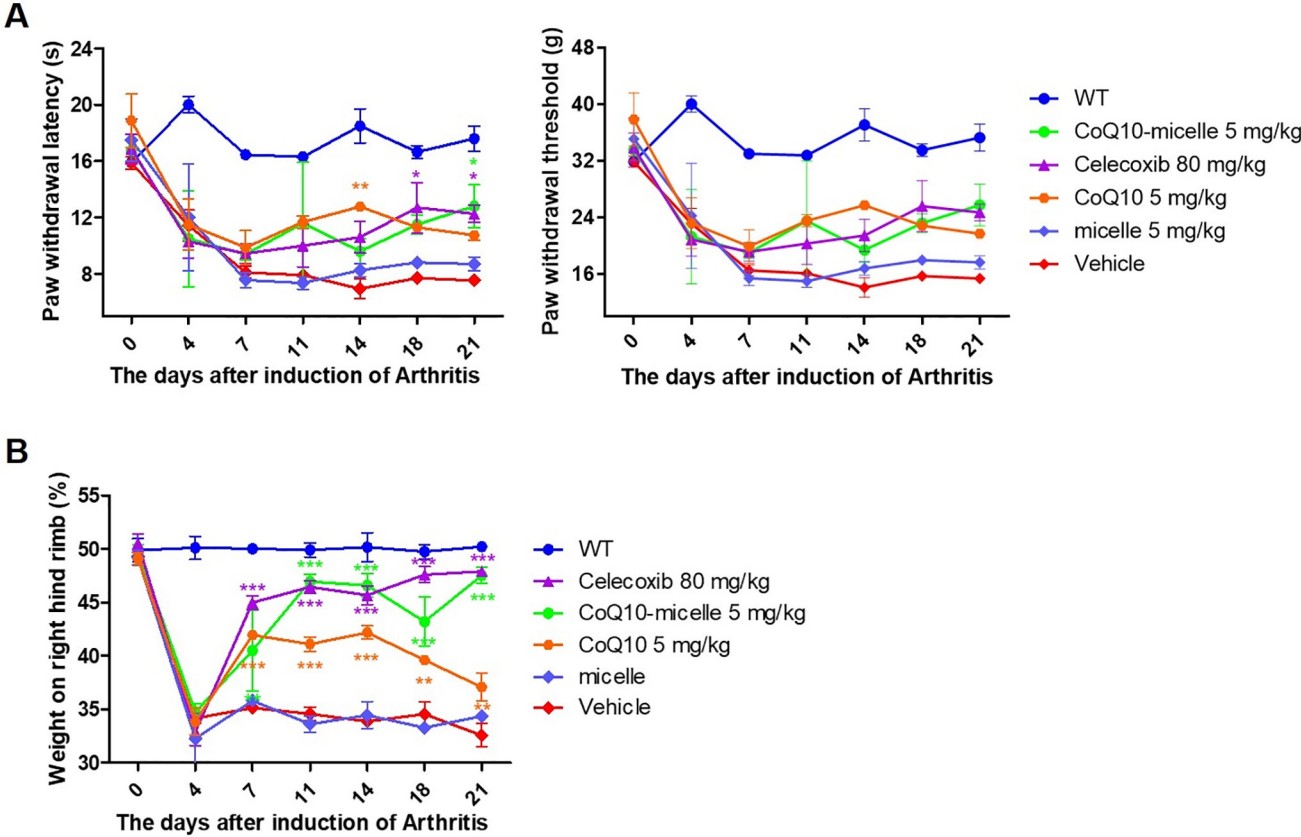

**Fig 2. The therapeutic effects of CoQ10-micelles on OA progression.** Animals were injected with MIA to induce OA. 3 days after induction, CoQ10-micelles, CoQ10, micelle, celecoxib were administered orally once a day. Animals were monitored for 21 days. (**A**) The PWL (left) and PWT (right) were used to measure pain in the indicated group (N = 3 per group) until day 21. (**B**) Weight-bearing was analyzed until day 21. Data are shown as means ± S.E.Ms. Statistical significance was assessed using the Bonferroni test. $^*p < 0.05$, $^{***}p < 0.005$.

reduced in the CoQ10-micelle group compared to the vehicle group (Fig 4A and 4B). Thus, CoQ10-micelles slowed OA progression.

## CoQ10-micelles reduced the levels of inflammatory mediators and catabolic factors in the synovium of MIA-induced OA rats

To explore whether CoQ10-micelles affected the expression of inflammatory mediators and catabolic factors involved in OA progression, we immunochemically stained tissue samples for IL-1β, IL-6, and MMP13. The IL-1β and IL-6 levels were reduced in the CoQ10-micelle group compared to the vehicle group (Fig 5A and 5B). The level of the catabolic factor MMP13 was also decreased in the CoQ10-micelle group compared to the vehicle group (Fig 5A and 5B). CoQ10-micelles also reduced the levels of mRNAs encoding catabolic factors in human OA chondrocytes (Fig 7A). Thus, CoQ10-micelles protected against OA progression by inhibiting inflammation and the catabolic response.

## CoQ10-micelle administration reduced inflammatory cell death

Inflammatory cell death is well known to cause tissue destruction during OA progression. To explore any role for CoQ10-micelles in this context, we immunohistochemically stained tissue

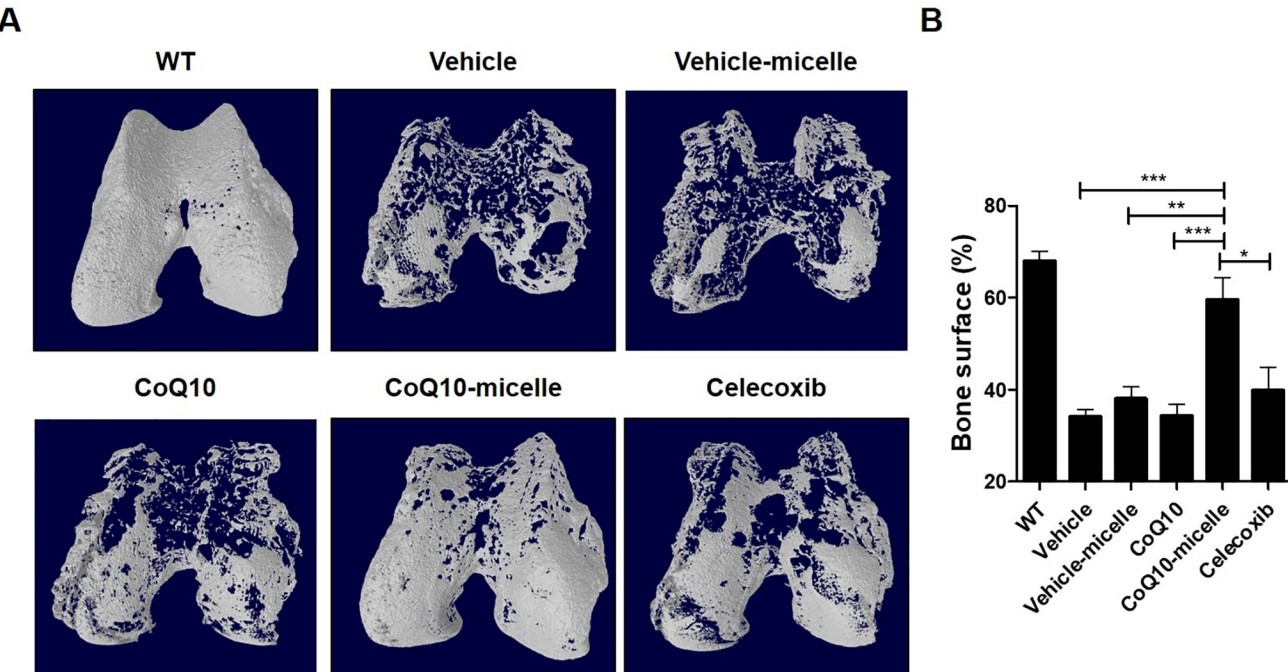

**Fig 3. CoQ10-micelles reduced bone erosion. (A)** Micro-CT was performed to measure bone loss in a WT group, a vehicle group, a CoQ10-micelle group, and a celecoxib group (N = 3 per group). **(B)** The bar graph shows the average bone surface density percentages. Data are shown as means ± S.E.Ms. Statistical significance was assessed using the Bonferroni test. **p < 0.01.

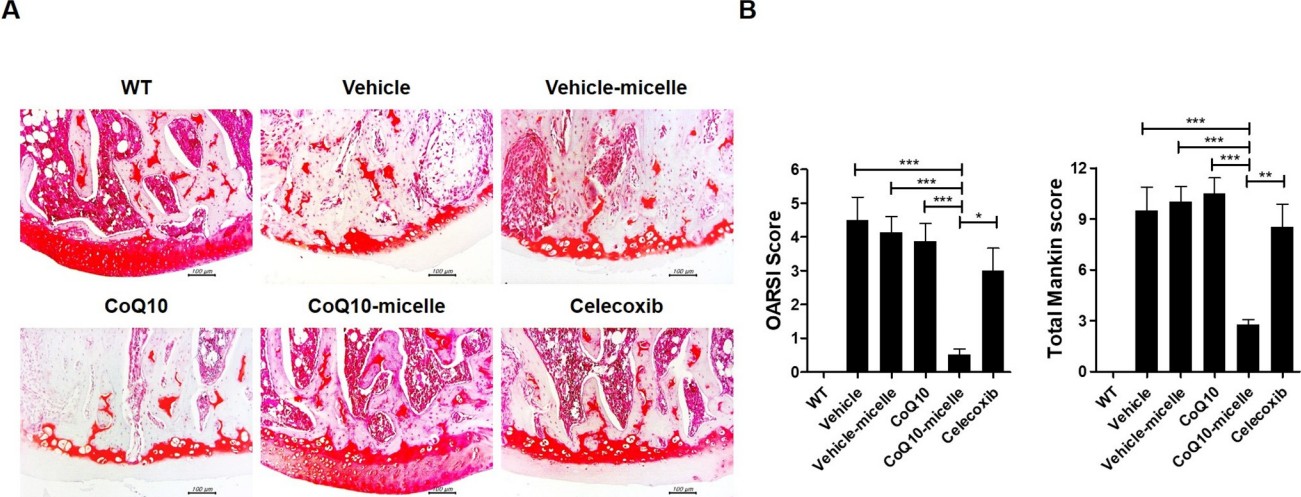

**Fig 4. CoQ10-micelles protected against cartilage destruction during OA progression. (A)** Safranin-O staining was used to evaluate cartilage destruction in a WT group, a vehicle group, a CoQ10-micelle group, and a celecoxib group (N = 3 per group). **(B)** The bar graphs show the average OARSI (left) and Makin (right) scores. Data are shown as means ± S.E.Ms. Statistical significance was assessed using the Bonferroni test. *p < 0.05.

samples for cell death markers including RIP1, RIP3, and phosphorylated-MLKL (pMLKL). The expression of all markers was lower in the CoQ10-micelle group than the vehicle group (Fig 6A and 6B). CoQ10-micelles also reduced the levels of mRNAs encoding inflammatory cell death markers in human OA chondrocytes (Fig 7B).

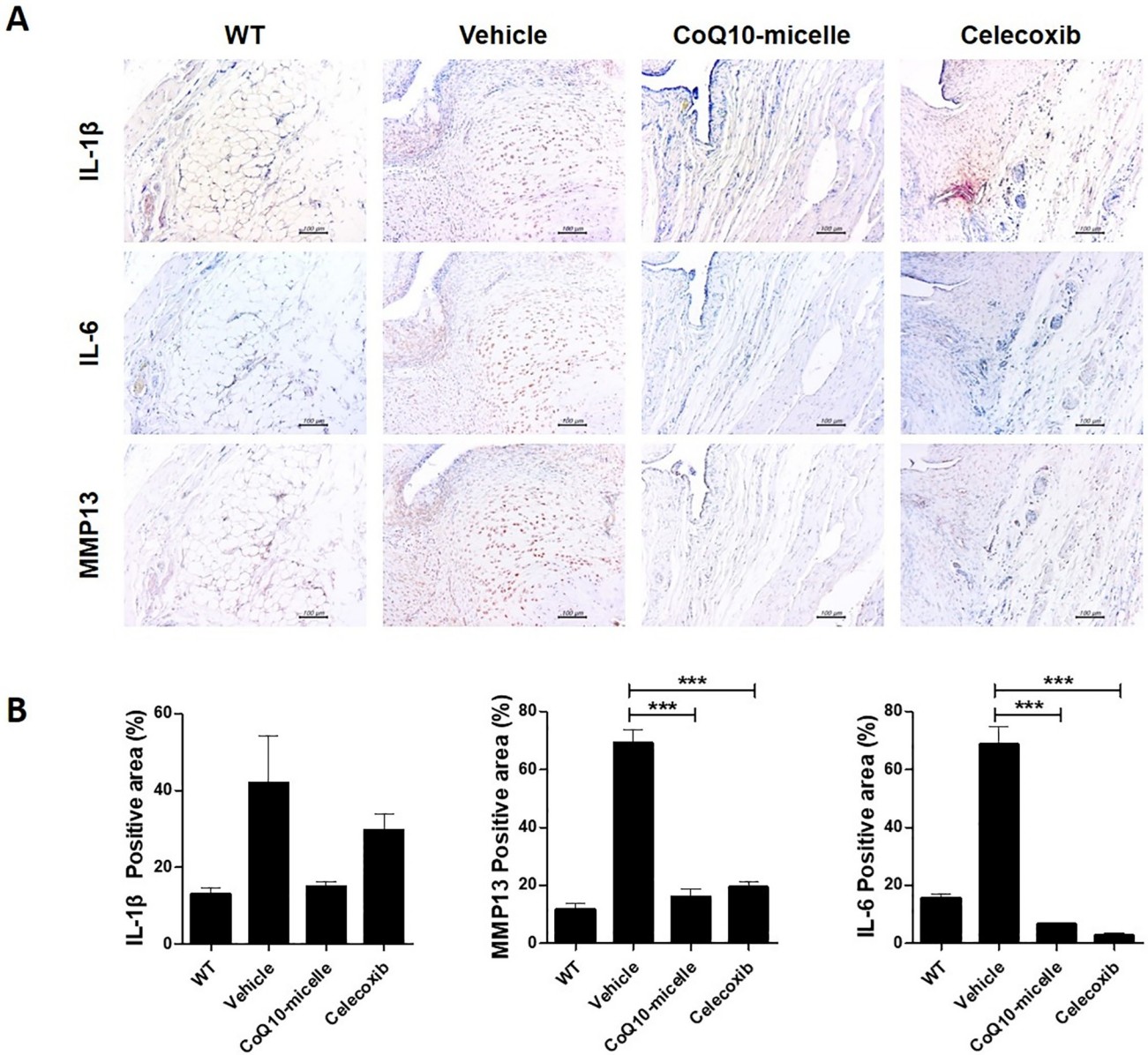

**Fig 5. The anti-inflammatory effects of CoQ10-micelles during OA progression. (A)** Representative images of immunohistochemical staining of IL-1β, IL-6, and MMP13 in the joint synovia of a WT group, a vehicle group, a CoQ10-micelle group, and a celecoxib group. **(B)** The bar graphs show the average positive areal percentages for IL-1β, IL-6, and MMP13. Data are shown as means ± S.E.Ms. Statistical significance was assessed using the Bonferroni test. *p < 0.05.

## Discussion

Osteoarthritis (OA) is the most common form of degenerative arthritis associated with pain and cartilage destruction in older adults. The pathogenesis includes inflammation. There is no cure; however, analgesic and anti-inflammatory medicines, such as corticosteroids and nonsteroidal anti-inflammatory drugs (NSAIDs) are available [31–33]. Several studies have shown that CoQ10 slows OA progression [17, 18, 22]. Although anti-oxidant and anti-inflammatory effects of CoQ10 have been reported [34–38], little is known about the effects of micellized-

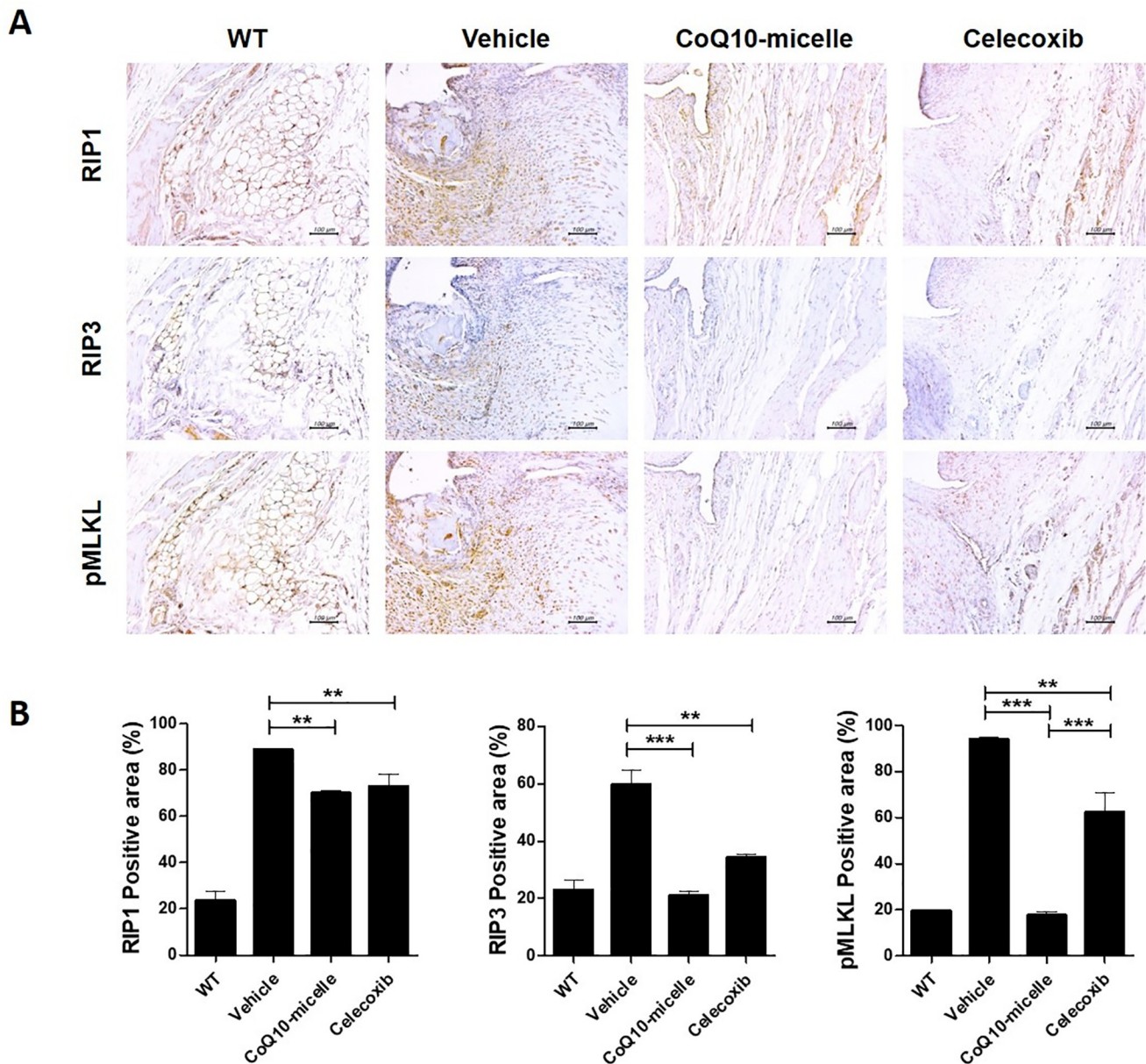

**Fig 6. The effect of CoQ10-micelles on inflammatory cell death during OA progression.** (A) Representative images of immunohistochemical staining of RIP1, RIP3, and pMLKL in the joint synovia of a WT group, a vehicle group, a CoQ10-micelle group, and a celecoxib group. (B) The bar graphs show the average positive areal percentages for RIP1, RIP3, and pMLKL. Data are shown as means ± S.E.Ms. Statistical significance was assessed using the Bonferroni test. $^*p < 0.05$.

CoQ10 on OA. We found that micellized-CoQ10 (CoQ10-micelles) inhibited OA development and progression.

During OA, the activation of catabolic factors triggers bone and cartilage degradation, and pain. MMPs are catabolic mediators which degrade extracellular matrix proteins, and cause inflammatory diseases such as OA. In OA, MMPs degrade cartilage and thus exacerbate OA. MMP activation and overexpression induces tissue destruction. Increased MMP levels were observed in patients with OA and experimental animals with OA [39]. MMP levels are

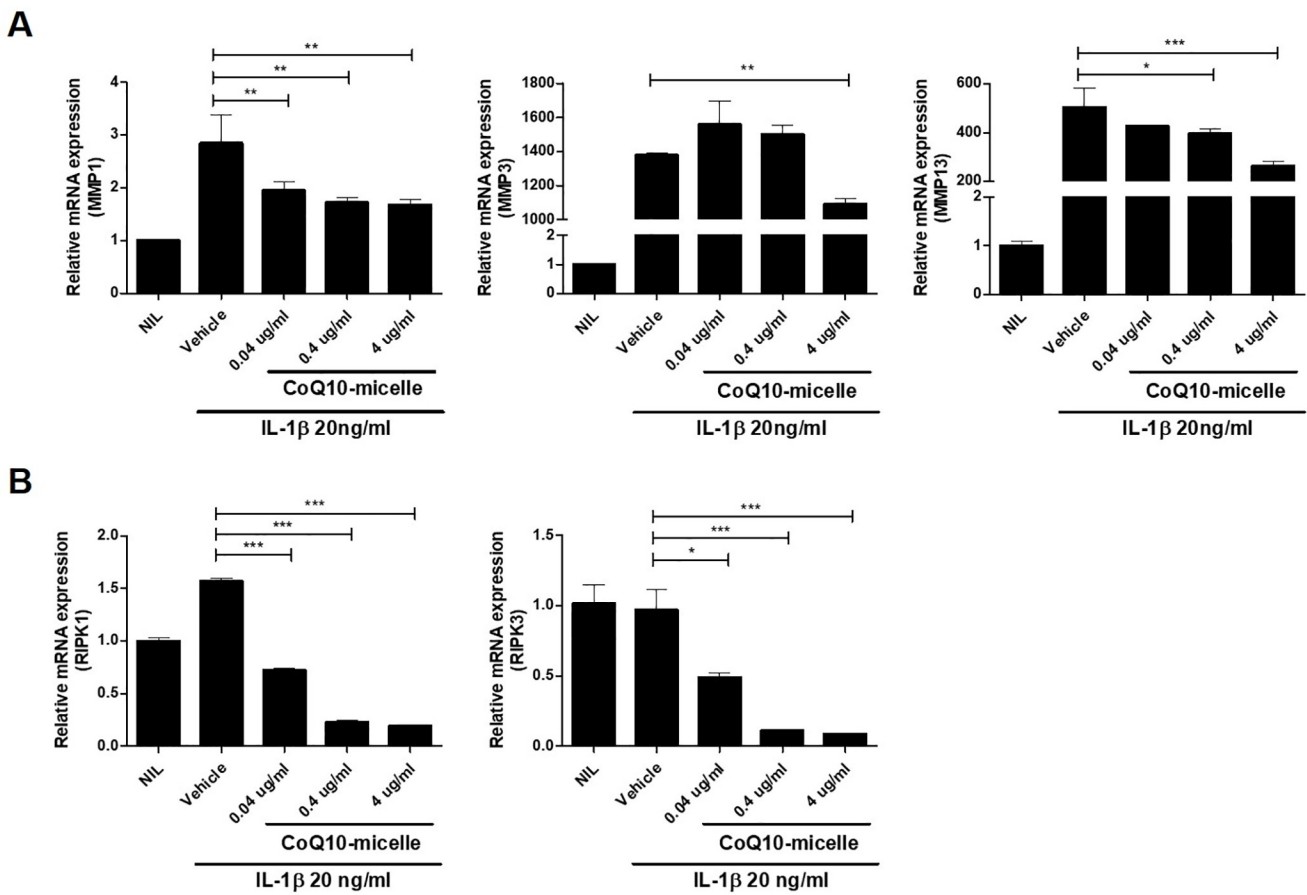

**Fig 7. The effects of CoQ10-micelles on the expression levels of catabolic and necroptotic factors in chondrocytes.** Chondrocytes were stimulated with IL-1β (20 ng/mL) in the absence or presence of CoQ10-micelles (0.04, 0.4, or 4 μg/mL) for 24 h, harvested, and mRNA extracted. **(A)** The bar graphs show the levels of mRNAs encoding MMP1, MMP3, and MMP13 in chondrocytes. **(B)** The levels of mRNAs encoding RIPK1 and RIPK3 in chondrocytes. Data are shown as means ± S.E.Ms. Statistical significance was assessed using the Bonferroni test. $^*p < 0.05$, $^{**}p < 0.01$, $^{***}p < 0.005$.

regulated by IL-1β, which is a key mediator of the inflammatory response. IL-1β is well-known to play a crucial role in OA [40]. Chondrocytes are the only cells found in cartilage; the cells control the structure of the extracellular cartilage matrix. Chondrocyte inflammation induced by IL-1β triggers cartilage destruction. We previously reported that MMP and IL-1β levels were increased in an OA animal model exhibiting such destruction [41–44].

CoQ10 exhibits clinical anti-oxidant and anti-inflammatory effects [35, 45]. CoQ10 deficiency has been associated with various diseases and five clinical phenotypes, thus encephalo-myopathy, severe infantile multisystemic disease, cerebellar ataxia, nephropathy, and isolated myopathy [46]. Previously, we reported that CoQ10 was therapeutic in patients with autoimmune diseases [20, 47]. CoQ10 ameliorated OA symptom in animal mode by regulating inflammatory cytokines [21]. Chang *et al.* reported that CoQ10 is the key factor and therapeutic target for the patient with OA [17]. Drug encapsulation affords several benefits. Drug side-effects are reduced [26]. Although micellized-CoQ10 has been used to treat certain diseases, OA was not among the diseases.

Here, we investigated whether micellized-CoQ10 (CoQ10-micelles) affected OA progression compared to CoQ10. We found that CoQ10-micelles showed better chondroprotective effect than CoQ10 in OA rats. Pain severity, bone erosion, and cartilage destruction were

significantly decreased by CoQ10-micelles treatment of experimental rats. Micro-CT and saf-ranin-O staining showed that bone erosion and cartilage destruction were reduced by CoQ10-micelles. OA-induced increases in catabolic mediators and MMPs were reduced by CoQ10-micelles. The levels of pro-inflammatory cytokines, including IL-1β and IL-6, were also significantly decreased; the drug exhibited an anti-inflammatory activity. The role played by inflammatory cell death (both pyroptosis and necroptosis) in OA pathogenesis is well defined. Several studies have reported increased inflammatory cell death during OA progression [48, 49]. Phosphorylation of MLKL by the protein kinase RIPK3 induces necroptosis and OA progression [50]. We found that CoQ10-micelles reduced the expression of RIP1, RIP3, and phosphorylated MLKL in synovial tissue. Next, we examined whether CoQ10-micelle show therapeutic effect in human chondrocytes. We found that CoQ10-micelle decreased mRNA level of MMP1, MMP3, MMP13, RIPK1, and RIPK3 in chondrocytes from OA patients. Our result demonstrated that CoQ10-micelle has therapeutic effect in not only animal model, but also in human.

## Conclusions

We explored whether CoQ10-micelles inhibited OA development and progression. CoQ10--micelles reduced pain, bone erosion, and cartilage destruction. The levels of pro-inflammatory cytokines and catabolic factors were decreased in CoQ10-micelle-treated OA rats and human OA chondrocytes. The levels of inflammatory cell death markers also decreased in CoQ10-mi-celle treated OA rats. Together, our findings suggest that CoQ10-micelles can be better choice than CoQ10 in clinical use to treat OA; the inflammatory response is downregulated.

## Supporting information

**S1 File.**
(DOCX)

**S1 Checklist. The ARRIVE guidelines 2.0: Author checklist.**
(PDF)

## Author Contributions

**Conceptualization:** Mi-La Cho, Sang J. Chung, Sung-Hwan Park.

**Data curation:** Hyun Sik Na, Jeong Su Lee, In Gyu Um, Keun-Hyung Cho.

**Formal analysis:** Hyun Sik Na.

**Investigation:** Hyun Sik Na, Ju Hwan Kim.

**Resources:** Ju Hwan Kim, Ga Hyeon Kim.

**Supervision:** Mi-La Cho, Sang J. Chung, Sung-Hwan Park.

**Validation:** Jin Seok Woo.

**Writing – original draft:** Jin Seok Woo.

**Writing – review & editing:** Jin Seok Woo, Mi-La Cho.

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
