## [Decision Letter · Decision Letter 0]

7 Apr 2022

PONE-D-22-05304Coenzyme Q10 encapsulated in micelles ameliorates osteoarthritis by inhibiting inflammatory cell deathPLOS ONE

Dear Dr. Cho,

Thank you for submitting your manuscript to PLOS ONE. After careful consideration, we feel that it has merit but does not fully meet PLOS ONE’s publication criteria as it currently stands. Therefore, we invite you to submit a revised version of the manuscript that addresses the points raised during the review process.

We look forward to receiving your revised manuscript.

Kind regards,

Rosanna Di Paola, MD

Academic Editor

PLOS ONE

Journal Requirements:

2. As part of your revision, please complete and submit a copy of the Full ARRIVE 2.0 Guidelines checklist, a document that aims to improve experimental reporting and reproducibility of animal studies for purposes of post-publication data analysis and reproducibility: https://arriveguidelines.org/sites/arrive/files/Author%20Checklist%20-%20Full.pdf (PDF). Please include your completed checklist as a Supporting Information file. Note that if your paper is accepted for publication, this checklist will be published as part of your article.

4. Please include a copy of Table 1 which you refer to in your text on page 10.

Reviewers' comments:

Reviewer's Responses to Questions

**Comments to the Author**

1. Is the manuscript technically sound, and do the data support the conclusions?

Reviewer #1: Partly

Reviewer #2: Partly

2. Has the statistical analysis been performed appropriately and rigorously? 

Reviewer #1: Yes

Reviewer #2: I Don't Know

3. Have the authors made all data underlying the findings in their manuscript fully available?

Reviewer #1: Yes

Reviewer #2: No

4. Is the manuscript presented in an intelligible fashion and written in standard English?

Reviewer #1: Yes

Reviewer #2: Yes

5. Review Comments to the Author

Reviewer #1: This manuscript demonstrates that delivery of Co-enzyme Q10 in micelles can alleviate pain, inflammatory cell death and tissue destruction in a rat model of osteoarthritis. In addition, they demonstrate that in vitro treatment Co-Q10 treatment attenuates the inflammatory cell death in OA patient chondrocytes induced by IL-1B.

There are issues that need to be addressed.

1. This manuscript does not provide new mechanisms of Co-enzyme Q10 in OA from what they published previously.

2. The authors state that micelles can increase the potency of drug delivery, this is not directly assessed in this manuscript. Administration of CoQ10 as a control would be needed to determine this.

3. They have used Celebrex as a positive treatment control, but it is unclear what they negative control mice are being given. Are they administered empty micelles?

4. In figure 5. It is unclear how the qPCR data was quantified for each gene of interest? Were these delta-delta Ct values or was the data relative RNA compared to a reference gene expression?

5. The primer sequences used should be provided.

Reviewer #2: With this work, the authors intend to demonstrate that micellized CoenzymeQ10 has a protective role on the OA inflammatory and catabolic features of the disease, using in the study a rat model of OA.

In general, the authors provide extensive several parameters that ensure the quality of the results obtained. The abstract and introduction is, in general written, in clear English with few flaws.

Nevertheless within the next sections Ia have serious concerns that should be clarified.

Materials and methods section:

Induction of osteoarthritis and treatment with CoQ10-micelles:

Suggestion: Please describe how many animals were considered for each group and how long after treatment with MIA were animals considered OA. Please describe also how long was the treatment with coQ10.

Results Section:

It would have been nice to see the improvement/decrease results expressed in concrete values of mean+/- SEM.

Legend of figure 2 needs improvement since it is not easy to understand what is stated in special in what refers to the timepoints in which the measures were taken.

Please explain in detail how you determined these areas of positivity for cytokines and MMP, in order to clarify the results presented, since the representative images are difficult to visualize in the quality presented.

Figure 6 and its legend do not correspond. Probably by mistake authors presented in figure 6 the same panels of representative immunuhistochemstry images for Cytokines and MMP from figure 5 and not representative images of immunohistochemical staining of RIP1, RIP3, and pMLKL as stated.

Since the effect of coQ10 in OA was already reported it would have been interesting to see if the levels of micellized co-Q10 would have increased the levels of the freed coQ10 in the blood samples/synovial fluid of the animals in study.

Although the English used is grammatically correct the text is sometimes confusing, especially in the discussion section.

Please reformulate the second paragraph of the discussion section.

E.g: “Increased MMP levels cause inflammatory diseases such as OA”, please explain.

The discussion fails to bridge more often between the results obtained by the authors in this model and with micellized co-Q10, and what is already known from previous studies with Q10 and OA. They also fail to mention at any point why human chondrocytes are used in the process and that the entire study was performed in a rat model.

In my opinion, although interesting, the study does not provide significant novelty and needs some revision before it can be published.

6. PLOS authors have the option to publish the peer review history of their article (what does this mean?). If published, this will include your full peer review and any attached files.

Reviewer #1: **Yes: **Robert Axtell

Reviewer #2: No

---

## [Author Response · Author response to Decision Letter 0]

11 May 2022

We really appreciate your time and effort to edit our manuscript. In this revised manuscript, we have resolved most of the issues raised by the reviewers as you can see in our response to their comments below. We think the reviewer’s and the editor’s suggestions have tremendously improved the quality of this manuscript and we hope the revised manuscript fulfils the requirements for its publication in Plos One. Detailed responses the reviewer’s critique are elaborated below.

---

## [Decision Letter · Decision Letter 1]

9 Jun 2022

Coenzyme Q10 encapsulated in micelles ameliorates osteoarthritis by inhibiting inflammatory cell death

PONE-D-22-05304R1

Dear Dr. 

We’re pleased to inform you that your manuscript has been judged scientifically suitable for publication and will be formally accepted for publication once it meets all outstanding technical requirements.

Kind regards,

Rosanna Di Paola, MD

Academic Editor

PLOS ONE

Additional Editor Comments (optional):

Reviewers' comments:

Reviewer's Responses to Questions

**Comments to the Author**

1. If the authors have adequately addressed your comments raised in a previous round of review and you feel that this manuscript is now acceptable for publication, you may indicate that here to bypass the “Comments to the Author” section, enter your conflict of interest statement in the “Confidential to Editor” section, and submit your "Accept" recommendation.

Reviewer #3: All comments have been addressed

2. Is the manuscript technically sound, and do the data support the conclusions?

Reviewer #3: Partly

3. Has the statistical analysis been performed appropriately and rigorously? 

Reviewer #3: Yes

4. Have the authors made all data underlying the findings in their manuscript fully available?

Reviewer #3: Yes

5. Is the manuscript presented in an intelligible fashion and written in standard English?

Reviewer #3: Yes

6. Review Comments to the Author

Reviewer #3: General comments:

This paper certainly has an excellent topic and very promising premises for future research. Despite this premise, it needs content revisions in order to be suitable for publication.

Keywords:

It would be advisable to change the keywords; some of these are also present in the paper's title.

It would be opportune in indicate the full addresses of the manufacturers from whom the materials used for conducting the experiments are procured, please standardize the whole section accordingly.

The authors need to clearly specify the exact number of animals used for all procedures; it is not easy to tell from the text how many animals were used in how many experimental groups and for how many analyses. Fix everything by taking a cue from this paper DOI: 10.3390/biom12040564 and cite it.

Lines 154-159: It would be desirable to improve the quality of the description of the methods used as well,I suggest taking a cue for the exposition of the PCR procedure from the following manuscript and insert the appropriate citation : DOI: 10.3390/life12010128

7. PLOS authors have the option to publish the peer review history of their article (what does this mean?). If published, this will include your full peer review and any attached files.

Reviewer #3: No

---

## [Editor Report · Acceptance letter]

15 Jun 2022

PONE-D-22-05304R1 

Coenzyme Q10 encapsulated in micelles ameliorates osteoarthritis by inhibiting inflammatory cell death 

Dear Dr. Cho:

I'm pleased to inform you that your manuscript has been deemed suitable for publication in PLOS ONE. Congratulations! Your manuscript is now with our production department. 

Kind regards, 

on behalf of

Dr. Rosanna Di Paola 

Academic Editor

PLOS ONE